# Risk factors for surgical site infections using a data-driven approach

J. M. van Niekerk[1,2,3], M. C. Vos[3], A. Stein[2], L. M. A. Braakman-Jansen[1]*, A. F. Voor in 't holt[3], J. E. W. C. van Gemert-Pijnen[1]

**1** Department of Psychology, Health and Technology/Centre for eHealth Research and Disease Management, Faculty of Behavioural Sciences, University of Twente, Enschede, The Netherlands, **2** Department of Earth Observation Sciences, Faculty of Geo-Information Science and Earth Observation (ITC), University of Twente, Enschede, The Netherlands, **3** Department of Medical Microbiology and Infectious Diseases, Erasmus MC University Medical Centre, Rotterdam, The Netherlands

* l.m.a.braakman-jansen@utwente.nl

## Abstract

### Objective

The objective of this study was to identify risk factors for surgical site infection from digestive, thoracic and orthopaedic system surgeries using clinical and data-driven cut-off values. A second objective was to compare the identified risk factors in this study to risk factors identified in literature.

### Summary background data

Retrospective data of 3 250 surgical procedures performed in large tertiary care hospital in The Netherlands during January 2013 to June 2014 were used.

### Methods

Potential risk factors were identified using a literature scan and univariate analysis. A multivariate forward-step logistic regression model was used to identify risk factors. Standard medical cut-off values were compared with cut-offs determined from the data.

### Results

For digestive, orthopaedic and thoracic system surgical procedures, the risk factors identified were preoperative temperature of ≥38˚C and antibiotics used at the time of surgery. C-reactive protein and the duration of the surgery were identified as a risk factors for digestive surgical procedures. Being an adult (age ≥18) was identified as a protective effect for thoracic surgical procedures. Data-driven cut-off values were identified for temperature, age and CRP which can explain the SSI outcome up to 19.5% better than generic cut-off values.

### Conclusions

This study identified risk factors for digestive, orthopaedic and thoracic system surgical procedures and illustrated how data-driven cut-offs can add value in the process. Future

**Data Availability Statement:** The medical ethical technical committee of Erasmus MC did not grant permission to publish these data due to ethical

considerations and the sensitivity of the data. Data are however available from the Erasmus MC upon reasonable request. Contact person: Dr. Juliëtte Severin, Infection prevention and control (IPC) and antimicrobial resistance (AMR) (Data Access) E-mail: info.microbiologie. infectieziekten@erasmusmc.nl.

**Funding:** This research was supported by the INTERREG V A (202085) funded project EurHealth-1Health (http://www.eurhealth1health.eu), part of a Dutch-German cross-border network supported by the European Commission, the Dutch Ministry of Health, Welfare and Sport, the Ministry of Economy, Innovation, Digitalisation and Energy of the German Federal State of North Rhine-Westphalia and the Ministry for National and European Affairs and Regional Development of Lower Saxony. The funders had no role in study design, data collection and analysis, decision to publish, or preparation of the manuscript.

**Competing interests:** The authors have declared that no competing interests exist.

studies should investigate if data-driven cut-offs can add value to explain the outcome being modelled and not solely rely on standard medical cut-off values to identify risk factors.

## Introduction

Surgical site infections (SSI), as defined by the European Centre for Disease Prevention and Control (ECDC) [1], make up 19.6% of the total number of healthcare-associated infections (HAIs) in Europe. With an estimated 81 089 patients in Europe having an HAI on any given day, almost 16 000 people in Europe are suffering from some form of SSI at any given time [2]. The burden of SSI can be measured in terms of increased length of stay in hospital, additional (surgical) procedures required, increased morbidity and mortality, as well as in economic terms [3].

Risk factors relating to the patient, procedure and the environment alter the odds of an SSI occurring. Research has been done to identify risk factors for SSI with the aim to identify preventative actions to reduce the incidence rate of SSI [4–10]. Patient-related risk factors for SSI, such as obesity, diabetes, surgery duration and the American Society of Anaesthesiologists (ASA) score are risk factors for digestive system, thoracic and orthopaedic surgical procedures [11–22]. Risk factors in low-income countries also include unemployment and level of education due to the disparity in socioeconomic status [14]. Risk factors can be modifiable or non-modifiable [23]. Modifiable risk factors are most interesting of the two since they can be changed preoperatively to reduce the risk of SSI.

The Segmentation of surgical procedures into homogenous groups makes it possible to find useful and relevant risk factors unique to each segment. Digestive system surgical procedures are more prone to SSI as they are generally clean-contaminated or dirty surgeries which make deep space SSI more likely. The occurrence of SSI after thoracic and orthopaedic surgeries are both relatively low because they are both typically clean surgeries, but the probability of attracting a deep space SSI after thoracic surgery is much higher compared to orthopaedic surgeries [15]. Because of these differences, we focus on digestive system, thoracic and orthopaedic surgical procedures for this study.

Multivariate logistic regression is the most common statistical model used to identify risk factors in longitudinal study design data [16]. Not all studies report the discriminatory power of the multivariate logistic regression model fitted. Risk factor identification studies do not usually specify how continuous variables cut-offs are determined. Cut-off values for variables such as age ($\geq$18) or patient temperature (37˚C) may seem intuitive or standard for clinical practice, but they may not statistically be the best cut-offs values determined by the data [17].

The objective of this study is to identify risk factors for SSI from digestive, thoracic and orthopaedic system surgeries using clinical and data-driven cut-off values. A second objective is to compare the identified risk factors in this study to risk factors identified in the literature.

## Materials and methods

### Literature search

A literature search was performed to identify known risk factors for SSI associated with digestive system surgical procedures, thoracic surgery and orthopaedic procedures using the corresponding medical subject headings (MeSH) linked data representation and the MEDLINE database.

Search strings used for MEDLINE literature search:

1. "Surgical Wound Infection"[Mesh] AND "Risk Factors"[Mesh] AND "Digestive System Surgical Procedures"[Mesh]

2. "Surgical Wound Infection"[Mesh] AND "Risk Factors"[Mesh] AND "Orthopaedic Procedures"[Mesh]

3. "Surgical Wound Infection"[Mesh] AND "Risk Factors"[Mesh] AND "Thoracic Surgery"[Mesh]

The search results were sorted, using the *Best Match* algorithm [18] developed by PubMed. Search results were deemed relevant using title and abstract screening. Risk factors were extracted if they were significant in a multivariable analysis until data saturation was achieved [19]. Risk factors identified, which were common to all three groups of surgeries, were defined as "general risk factors" in this study.

## Setting and data collection

The Erasmus MC University Medical Centre in Rotterdam is the largest university medical hospital in the Netherlands with more than 1 300 beds [15]. The data used for this study were anonymised in accordance with the Dutch Personal Data Protection Act (WBP). Approval from the Medical Ethical Research Committee was obtained (MEC-2018-1185).

A weekly prevalence survey was performed by infection control practitioners (ICP) from January 2013 until December 2013 and two-weekly until June 2014 using a semi-automated algorithm proposed by Streefkerk et al. [20, 21]. This algorithm was used to calculate a nosocomial infection index (NII) which was then verified by ICP in case of a positive outcome to determine whenever an HAI was present or not. An ICP verified all patients with an $NII > 7$, and a definite SSI outcome was concluded by the ICP using the electronic patient data system. This outcome was used in this study as the occurrence of SSI outcome variable.

Data were extracted from a centralised database, containing cross-departmental data, clinical synopsis reports, infectious disease consultation reports, laboratory results and imaging reports. Data regarding the prescription of antimicrobials, in the J01 class of the Anatomical Therapeutic Chemical (ATC) classification system [22], were also included. Surgeries were included if they were part of the three groups of surgeries under investigation in this study and had a point prevalence measurement within 30 days after the surgery took place. If a second surgery took place within 30 days after an included surgery, then the recent surgery was excluded. All emergency surgeries were excluded to avoid possible undesirable confounding effects relating to the urgency and necessity of the surgeries.

## Statistical analysis

The differences in the averages of variables with missing values and those without were evaluated using t-tests and were found statistically significant. These tests, together with Little's MCAR test, convinced us that the missing values were not completely randomly missing and that we could not make use of more simple imputation methods. Therefore, we chose to use conditional Markov chain Monte Carlo (MCMC) with multiple imputations for the imputation process [24, 25].

Two methods were used to discretise continuous measurement variables: 1) standard medical cut-offs as used by Erasmus MC and 2) recursive partitioning [17]. Recursive partitioning is a data-driven, supervised discretisation method, used to group continuous values with similar outcomes optimally. The data-driven method was used to test and confirm if the standard medical cut-offs were the best way to explain the outcome variable for the groups of surgical procedures considered.

To build a prognostic prediction model for SSI, Hosmer et al. suggest fitting a univariate logistic regression model to each variable separately and if the p-value is less than a specific p-value, 0.1 is this case, then consider the variable good enough to include in the multivariate logistic regression model [26]. A univariate analysis was performed for each of the three groups of surgeries using the variables identified from the literature search. Significant variables ($p < 0.1$) in the univariate analysis were added to the list of variables associated with each group of surgery, together with the variables identified from the literature search. This resulted in an extended list of general risk factors as more risk factors were common across the three groups of surgeries.

A multivariate logistic regression model was built using a forward stepwise approach for each of the three groups of surgeries [27]. The general risk factors were first added to the model and then the risk factors unique to each surgery group in the order of the Akaike information criterion (AIC) until convergence was reached. In this case, we chose the conversion of the model to imply that there are no additional variables which can be added which will be statistically significant with a p-value of less than 0.05 or an AIC of 3.8415. Model performance was determined using the Gini coefficient after each step of the multivariate model, and the difference is reported as the marginal contribution of surgery group-specific risk factors for this study [19, 28]. Model performance was cross-validated using 5-fold cross-validation to estimate how the model would perform on new data [29]. R [30] was used in this study together with packages mice (multiple imputation) [31], smbinning (recursive partitioning) [32], dplyr (data wrangling) [33], finalfit (formatting of tables) [34] and scorecard (cross-validation) [35].

Approval was obtained from the Medical Ethical Committee of Erasmus MC (MEC-2018-1185) to perform this study. Data were analysed anonymously, and thus no further consent was obtained.

## Results

### Literature search

The literature search resulted in 1 422 research papers (as at 5 March 2020) using the MeSH headings in the PubMed search engine. We identified 24 research papers, published from 2008 until 2019, which contained statistically significant results from a multivariate analysis. A total of 79 risk factors were identified for the three groups of surgical procedures [11–13, 16, 23, 36–54] (S1 Table). Age, ASA class, body mass index (BMI), preoperative length of stay and diabetes were identified as general risk factors from the literature search. In total, 29 risk factors for digestive system surgical procedures, 31 for orthopaedic procedures and 19 for thoracic surgeries were identified. This amounted to 59 unique risk factors, of which 15 were present in more than one group of surgeries.

### Risk factor identification

A total of 21 of the 59 unique risk factors could be replicated using our own data. The variable describing the type of surgery was used to create three homogenous groups of surgical procedures. The emergency classification variable was used to exclude emergency surgeries from the study such that 19 risk factors remained (Table 1). We observed 3 250 surgeries over the study period and excluded 526 (16.2%) emergency surgeries to be left with 2 724 surgical observations. CRP and temperature data were available for 52.55% (60.47% for in-patients) and 96.88% of all surgeries respectively.

The significant univariate results of digestive system, orthopaedic and thoracic surgical procedures are shown in Table 2. Antibiotic use, CRP and temperature were added to the list of

**Table 1. Variable names and definitions used to investigate the occurrence of SSI in this study.**

| Variable | Surgery group | Definition |
|---|---|---|
| Demographic | | |
| Gender | D,O | Gender of patient (Male/Female) |
| Age | D,O,T | Age of patient on the day of surgery (Years) |
| ASA class | D,O,T | ASA class of patient (I-V) |
| BMI | D,O,T | BMI of patient at the time of surgery. |
| Behavioural | | |
| Alcohol use | O | Alcohol use of patient at the time of surgery (Current/Never/Past). |
| Smoking | D,O | Smoking status of patient at the time of surgery (Current/Never/Past). |
| Comorbidities | | |
| Heart disease | O,T | Patient has a history of heart disease at the time of surgery (Yes/No). |
| Liver disease | D | Patient has a history of liver disease at the time of surgery (Yes/No). |
| Hypertension | O | Patient has a history of hypertension (Yes/No). |
| Diabetes | D,O,T | Patient has diabetes Type I or II at the time of surgery (Yes/No). |
| Measurement | | |
| Temperature | D | Highest temperature of patient in the past 7 days before surgery. |
| CRP | O | Highest CRP of patient in the 7 days before surgery. |
| Leukocyte | D | Highest leukocyte level of patient in the 7 days before surgery. |
| Serum total protein | D | Highest serum total protein of patient in the 7 days before surgery. |
| Glucose | D | Highest glucose level of patient in the 7 days before surgery. |
| Haemoglobin | D | Highest haemoglobin level of patient in the 7 days before surgery. |
| Operative | | |
| Preoperative length of stay | D,O,T | Preoperative length of hospital stay of patient at the time of surgery (Days). |
| Antibiotic use | T | Antibiotic (WHO ATC code J01 [22]) use of patient at the time of surgery (Yes/No). |
| Duration of surgery | D,O | Duration of the surgical procedure (Minutes). |

D, Digestive system surgical procedures; O, Orthopaedic system surgical procedures; T, Thoracic system surgical procedures; ASA, American Society of Anaesthesiologists; CRP, C-reactive protein; BMI, Body Mass Index; SSI, Surgical Site Infection; ATC, Anatomical Therapeutic Chemical; WHO, World Health Organization.

general risk factors after being found statistically significant in the univariate analysis–increasing the number of general risk factors to 8. Diabetes was identified as a general risk factor from our literature search but was not found significant in any of the three univariate analyses in our own study. For digestive system surgical procedure and thoracic procedures, the data-driven cut-off for age was obtained as 23 years and both the standard cut-off (18 years) and the data-driven cut-off were statistically significant with p-values of less than 0.001 which resulted in rejecting the null hypothesis that the coefficient associated with the age of the patient is zero. For orthopaedic procedures, the data-driven cut-off for the temperature (39 degrees) was found statistically significant, but the standard medical cut-off not. A data-driven CRP cut-off of 8.1 was identified for orthopaedic surgical procedures as opposed to a standard medical CRP cut-off of 10; both cut-offs are statistically significant.

The multivariate results using standard medical cut-offs and data-driven cut-offs are shown in Tables 3 and 4, respectively. The temperature variable was statistically significant in the multivariate analysis using the data-driven cut-offs for all three groups of surgeries, but not in one of the multivariate analysis using the medical standard cut-offs. The duration of the surgery was the only statistically significant variable in the multivariate analyses which was not

**Table 2. Digestive system surgical procedures: univariate analysis of risk factors for the future occurrence of SSI.**

| Variable | | SSI = No (2 600) | SSI = Yes (124) | Univariate OR (95%CI, P-value) |
|---|---|---|---|---|
| Digestive System Surgical Procedures | | | | |
| Gender | Female | 359 (43.9)[2] | 24 (33.8) | Reference |
| | Male | 458 (56.1) | 47 (66.2) | 1.54 (0.93–2.60, p = 0.099) |
| Age[1] | ≤18 | 246 (30.1) | 8 (11.3) | Reference |
| | >18 | 571 (69.9) | 63 (88.7) | 3.39 (1.70–7.77, p<0.001) |
| Age (data-driven) | ≤23 | 258 (31.6) | 8 (11.3) | Reference |
| | >23 | 559 (68.4) | 63 (88.7) | 3.63 (1.82–8.32, p<0.001) |
| Antibiotic use | No | 496 (60.7) | 17 (23.9) | Reference |
| | Yes | 321 (39.3) | 54 (76.1) | 4.91 (2.85–8.86, p<0.001) |
| Temperature[1] | ≤36.5 | 0 (0.0) | 0 (0.0) | NA |
| | (36.5,37.5] | 98 (12.0) | 2 (2.8) | Reference |
| | >37.5 | 719 (88.0) | 69 (97.2) | 4.70 (1.44–28.91, p = 0.033) |
| Temperature (data-driven) | ≤38 | 535 (65.5) | 20 (28.2) | Reference |
| | (38,39] | 187 (22.9) | 25 (35.2) | 3.58 (1.95–6.66, p<0.001) |
| | >39 | 95 (11.6) | 26 (36.6) | 7.32 (3.94–13.79, p<0.001) |
| CRP[1] | ≤10 | 397 (48.6) | 21 (29.6) | Reference |
| | >10 | 420 (51.4) | 50 (70.4) | 2.25 (1.35–3.89, p = 0.003) |
| CRP (data-driven) | ≤8.1 | 365 (44.7) | 18 (25.4) | Reference |
| | >8.1 | 452 (55.3) | 53 (74.6) | 2.38 (1.39–4.24, p = 0.002) |
| Preoperative length of stay (Days) | Mean Days (SD) | 6.6 (24.1) | 12.1 (37.3) | 1.01 (1.00–1.01, p = 0.092) |
| Duration of surgery | Mean Minutes (SD) | 243.6 (143) | 330.4 (190.8) | 1.00 (1.00–1.01, p<0.001) |
| Orthopaedic Procedures | | | | |
| ASA class | ASA CLASS I | 196 (26.8) | 6 (33.3) | |
| | ASA CLASS II | 339 (46.4) | 6 (33.3) | 0.58 (0.18–1.87, p = 0.348) |
| | ASA CLASS III | 182 (24.9) | 4 (22.2) | 0.72 (0.18–2.55, p = 0.612) |
| | ASA CLASS ≥ IV | 13 (1.8) | 2 (11.1) | 5.03 (0.69–24.47, p = 0.062) |
| Alcohol use | Current | 327 (44.8) | 6 (33.3) | Reference |
| | Never | 339 (46.4) | 8 (44.4) | 1.29 (0.44–3.94, p = 0.645) |
| | Past | 64 (8.8) | 4 (22.2) | 3.41 (0.85–12.26, p = 0.063) |
| Antibiotic use | No | 591 (81.0) | 8 (44.4) | Reference |
| | Yes | 139 (19.0) | 10 (55.6) | 5.31 (2.06–14.16, p<0.001) |
| Temperature (data-driven) | ≤39 | 695 (95.2) | 14 (77.8) | Reference |
| | >39 | 35 (4.8) | 4 (22.2) | 5.67 (1.55–16.79, p = 0.003) |
| Thoracic Surgery | | | | |
| Age[1] | ≤18 | 232 (22.0) | 16 (45.7) | Reference |
| | >18 | 821 (78.0) | 19 (54.3) | 0.34 (0.17–0.67, p = 0.002) |
| Age (data-driven) | ≤23 | 226 (21.5) | 16 (45.7) | Reference |
| | >23 | 827 (78.5) | 19 (54.3) | 0.32 (0.16–0.65, p = 0.001) |
| BMI | Mean (SD) | 24.5 (5.3) | 22.1 (4.2) | 0.91 (0.85–0.98, p = 0.010) |
| Alcohol use | Current | 534 (50.7) | 11 (31.4) | Reference |
| | Never | 422 (40.1) | 18 (51.4) | 2.07 (0.98–4.57, p = 0.061) |
| | Past | 97 (9.2) | 6 (17.1) | 3.00 (1.01–8.09, p = 0.034) |
| Antibiotic use | No | 705 (67.0) | 18 (51.4) | Reference |
| | Yes | 348 (33.0) | 17 (48.6) | 1.91 (0.97–3.77, p = 0.060) |
| Temperature[1] | ≤36.5 | 0 (0.0) | 0 (0.0) | NA |
| | (36.5,37.5] | 302 (28.7) | 3 (8.6) | Reference |
| | >37.5 | 751 (71.3) | 32 (91.4) | 4.29 (1.52–17.94, p = 0.017) |

(*Continued*)

**Table 2.** (Continued)

| Variable | | SSI = No (2 600) | SSI = Yes (124) | Univariate OR (95%CI, P-value) |
|---|---|---|---|---|
| Temperature (data-driven) | ≤38 | 882 (83.8) | 20 (57.1) | Reference |
| | >38 | 171 (16.2) | 15 (42.9) | 3.87 (1.91–7.67, p<0.001) |
| CRP[1] | ≤10 | 684 (65.0) | 17 (48.6) | Reference |
| | >10 | 369 (35.0) | 18 (51.4) | 1.96 (1.00–3.88, p = 0.050) |
| Haemoglobin[1] | ≤8.6 | 665 (63.2) | 21 (60.0) | Reference |
| | (8.6,10.5] | 358 (34.0) | 11 (31.4) | 0.97 (0.45–2.00, p = 0.942) |
| | >10.5 | 30 (2.8) | 3 (8.6) | 3.17 (0.72–9.85, p = 0.074) |

CRP, C-reactive protein; OR, Odds Ratio; BMI, Body Mass Index; NA, Not Applicable; CI, Confidence Interval; SSI, Surgical Site Infection; OR, Odds ratio; Data-driven, cut-off values determined using recursive partitioning.

[1]Standard Erasmus MC clinical cut-offs.

[2]The percentage distribution of the SSI outcome is provided in brackets next to the frequency for each variable.

identified as a general risk factor to increase the odds of SSI by approximately 6% for every 30 minutes spent in surgery. For digestive surgical procedures, the addition of duration of surgery to the multivariate model increased the Gini coefficient from 0.46 to 0.52 based on standard medical cut-offs and from 0.57 to 0.62 for the multivariate model based on the data-driven cut-offs. This increase translates into a 12.5% and 8.8% increase in the Gini coefficient, respectively. Neither the orthopaedic nor the thoracic group of surgical procedures had any statistically significant risk factors which are not part of the general risk factors group of surgeries. The Gini coefficient of the data-driven multivariate model is 19.5% (0.62 vs 0.52) higher than the multivariate model based on the standard medical cut-offs. The 5-fold cross-validated 95% confidence intervals for the Gini coefficients based on the validation samples of the data-driven models are (0.49, 0.72) for digestive procedures, (0.21, 0.86) for orthopaedic procedures and (0.21,0.70) for thoracic procedures.

An overview of the study results (Table 5) shows that 10 of the 19 risk factors, identified during the literature search, were not statistically significant in the univariate or multivariate analysis for any of the surgery groups. BMI and diabetes were identified across all three groups of surgeries and multiple studies as risk factors for SSI but were not statistically significant in this study. Temperature and the duration of the surgery were confirmed as risk factors for digestive system surgeries, and similarly, antibiotic use and age were confirmed as risk factors

**Table 3. Multivariate analysis risk factors for the occurrence of SSI by group of surgeries using standard medical cut-offs.**

| Risk factor by surgery group[1] | Coefficient | Multivariate OR (95%CI) | P-value |
|---|---|---|---|
| Digestive System Surgical Procedures | | | |
| Antibiotic use | 1.240 | 3.455 (1.951–6.384) | <0.001 |
| Duration of surgery (Minutes) | 0.003 | 1.003 (1.001–1.004) | <0.001 |
| CRP >10 | 0.803 | 2.232 (1.302–3.951) | 0.004 |
| Orthopaedic Surgical Procedures | | | |
| Antibiotic use | 1.670 | 5.315 (2.059–14.158) | <0.001 |
| Thoracic Surgical Procedures | | | |
| Age >18 | -4.195 | 0.146 (0.058–0.351) | <0.001 |
| Antibiotic use | 1.311 | 4.849 (2.035–12.266) | <0.001 |

CRP, C-reactive protein; CI, Confidence Interval; OR, Odds ratio.

[1]The multivariate analysis was performed using Erasmus MC clinical cut-offs.

**Table 4. Multivariate analysis risk factors for the occurrence of SSI by group of surgeries using data-driven cut-offs.**

| Risk factor by surgery group[1] | Coefficient | Multivariate OR (95%CI) | P-value |
|---|---|---|---|
| Digestive System Surgical Procedures | | | |
| Temperature [38,39] | 1.067 | 2.907 (1.556–5.497) | <0.001 |
| Temperature >39 | 1.732 | 5.650 (2.952–10.947) | <0.001 |
| Antibiotic use | 1.201 | 3.322 (1.856–6.200) | <0.001 |
| Duration of surgery (Minutes) | 0.002 | 1.002 (1.001–1.004) | 0.003 |
| CRP >8.1 | 0.639 | 1.894 (1.062–3.510) | 0.035 |
| Orthopaedic Surgical Procedures | | | |
| Antibiotic use | 1.552 | 3.665 (1.370–10.006) | 0.009 |
| Temperature >39 | 1.224 | 5.120 (1.316–16.387) | 0.009 |
| Thoracic Surgical Procedures | | | |
| Age >17 | -1.847 | 0.158 (0.055–0.426) | <0.001 |
| Antibiotic use | 1.597 | 4.939 (1.896–14.043) | 0.002 |
| Temperature >38 | 0.824 | 2.280 (1.098–4.653) | 0.024 |

Data-driven, cut-off values determined using recursive partitioning; CRP, C-reactive protein; CI, Confidence Interval; OR, Odds ratio.

[1]The multivariate analysis was performed using data-driven cut-offs.

for thoracic surgeries. Antibiotic use and CRP were identified as risk factors for digestive surgeries from the multivariate analysis, which were identified during the literature search for thoracic and orthopaedic surgeries, respectively. Antibiotic use and temperature were

**Table 5. Statistical significance of risk factors and the source which lead them to be considered by surgical procedure.**

| Risk Factor | Significance[1] | Digestive System[2] | Orthopaedic[2] | Thoracic[2] |
|---|---|---|---|---|
| Age | $D_U, T_M$ | [38, 11, 43, 47] | [16] | [12] |
| Alcohol use | $O_U, T_U$ | | [51] | |
| Antibiotic use | $D_M, O_M, T_M$ | | | [40] |
| ASA Class | $O_U$ | [37, 39, 41, 43, 54] | [16, 51, 53] | [16] |
| BMI | None | [44] | [51–53] | [42] |
| CRP | $D_M$ | | [16] | |
| Diabetes | None | [38, 47, 50] | [16, 45, 51, 53] | [13] |
| Duration of surgery | $D_M$ | [36, 38, 41, 43, 44, 49, 54] | [16, 45, 51, 53] | |
| Gender | $D_U$ | [38, 11, 43] | [16, 51] | |
| Glucose | None | [47] | | |
| Haemoglobin | None | [11, 44, 54] | | |
| Heart Disease | None | | [51] | [12] |
| Hypertension | None | | [51] | |
| Leukocyte | None | [55] | | |
| Liver disease | None | [54] | | |
| Preoperative length of stay | $D_U$ | [41, 50] | [16, 52] | [12, 13, 40] |
| Serum total protein | None | [36, 49] | | |
| Smoking | None | [49] | [51–53] | |
| Temperature | $D_M, O_M, T_M$ | [55] | | |

D, Digestive system surgical procedures; O, Orthopaedic system surgical procedures; $_U$, Significant in univariate analysis; $_M$, Significant in multivariate analysis; T, Thoracic system surgical procedures; ASA, American Society of Anaesthesiologists; CRP, C-reactive protein; SSI, Surgical Site Infection; BMI, Body Mass Index.

[1]During which part of the analysis the risk factor was found statistically significant.

[2]References to the literature which had the risk factor as a multivariate result for each group of surgeries.

statistically significant for all three groups of surgeries and were included because of two studies regarding thoracic and digestive system surgeries, respectively [40, 55].

## Discussion

We identified temperature and antibiotics used at the time of surgery as risk factors for digestive, orthopaedic and thoracic system surgical procedures in this study. The duration of the surgery was identified as a risk factor for digestive surgical procedures. Being an adult (age ≥ 18) was identified as a protective effect for thoracic surgical procedures. Data-driven cut-offs were identified for temperature, CRP and age, which differ from the standard medical cut-offs. Temperature would not have been identified as a risk factor if only standard medical cut-offs were considered. From our literature search, we identified age, ASA class, BMI, preoperative length of stay and diabetes as general risk factors, while CRP, temperature and antibiotic use were identified as general risk factors because of this study.

The identified risk factors may be classified as modifiable or non-modifiable, depending upon the circumstances of the patient like the complexity of his condition. For instance, the temperature of a patient may be high because of an existing infection, which is why the surgery is needed in the first place and may not be modifiable before surgery. Age, on the other hand, may be a modifiable risk factor if the surgery can be postponed for several years, e.g. due to a heart defect. This study revealed that children are more likely to be diagnosed with an SSI after thoracic surgery than adults. There are studies which identify risk factors for children after thoracic surgeries, but none found that being a child is a risk factor for SSI [42, 48] after undergoing thoracic surgery. We segmented the thoracic surgeries between adults and children and obtained multivariate results for children and adults separately. The multivariate model based only on children (age ≤ 18) did not reveal any significant results, contrary to the results of the thoracic study which found age to be a risk factor for children [12]. This absence could be partly due to the small study population size of 248. Antibiotic usage was the only significant factor in the multivariate analysis of thoracic surgeries based on adults. The other two groups of surgical procedures were consistent in terms of their statistical significance of risk factors based on adults.

The data-driven cut-offs confirmed the existing standard medical cut-offs. On average the clinical cut-off for temperature was one degree Celsius lower, while for digestive system surgical procedures, the clinical cut-off for CRP (10) was just less than two units more than the data-driven cut-off of 8.1. This means that there is a greater difference between the occurrence of SSI for patients with a CRP below and above 8.1 than below and above 10. The data-driven cut-offs improved the ability of the statistical model to explain the occurrence of SSI. The performance of the digestive system surgical procedure prediction model increased by 19.5% due to using data-driven cut-offs rather than the standard medical cut-offs. Using data-driven cut-offs, we were able to identify temperature as a risk factor for all three groups of surgical procedures. If standard clinical cut-offs were used, temperature would not have been significant from the multivariate analysis. This potential oversight illustrates the importance of evaluating the cut-offs used for continuous variables against the data before identifying risk factors.

Antibiotic use, temperature and CRP were added to the list of general risk factors by incorporating the statistically significant results of the univariate analysis. These risk factors might have been overlooked when the focus was on only one type of surgery. Temperature was identified as a risk factor in the multivariate results for all three groups of surgical procedures, whereas the literature search identified it only for digestive surgeries. Antibiotic use was not found during our literature search for digestive or orthopaedic surgical procedures but was found significant for both groups of surgeries in the multivariate analysis of our study.

The Centres for Disease Control and Prevention (CDC), the European centre for disease prevention and control (ECDC), World Health Organisation (WHO) and Netherlands National Institute for Public Health and the Environment (RIVM) suggest maintaining normothermia intraoperatively to prevent undesirable hypothermia (during some thoracic and neurosurgeries, hypothermia may be desirable). [56–58] A lower intraoperative bound for temperature of 35.5˚C to 36˚C is explicitly mentioned, and only the RIVM mention an upper bound of 38˚C which is consistent with the risk factors identified in our study. An upper limit for preoperative temperature should, therefore, be investigated instead of only the lower limit. The four health organisations refer to the proper administration and timing of surgical antimicrobial prophylaxis, but not to the proper preoperative use of standard prescription antibiotics. Systemic antibiotics are typically prescribed to stabilise patients before undergoing surgery. A possible explanation for the increased occurrence of SSI associated with antimicrobials prescribed before surgery could be that these patients were not completely stabilised before surgery which increased their risk of SSI. The proper preoperative use of antibiotics should be well defined, and the reason why antibiotic-use was identified as a risk factor for SSI should be further investigated.

## Limitations

This is a retrospective, single-centre study, and therefore the data were not collected for the purpose of this study. Even though cross-validation was performed to estimate model performance on new data, the models were not externally validated. Surgeries were aggregated into three broad groups of surgical procedures which serve as a proxy for the reason for surgery but leads to the loss of information regarding the exact reasons for the surgery. Some measurements, like temperature and CRP, were not always present and was partly overcome using imputation. Patient information concerning smoking and drinking habits may be understated due to incomplete medical records. The literature search used for this study was not exhaustive but rather based on the principal on data saturation. A comprehensive list of variables related to the nutritional and immunological alterations of the patients was not included in the analyses as they were not available from the data. We used a 30-day outcome period in which we observe if an SSI was present or not, but according to the CDC definition, this outcome period should be one year for surgical implantation procedures. Since our data only spans over 18 months, it was not possible to use a 12-month outcome window for all surgical implantation procedures, which is a limitation of this study. The administration of prophylaxis and the optimal timing thereof is an important risk factor for the occurrence of SSI. However, these data were not available.

## Future work

Future work will investigate the modifiability of the risk factors identified in this study in more detail, as the circumstances under which this occurs are hitherto unclear. The exact purpose of the use of antibiotics over the time of surgery was not investigated in depth, which can be done in future studies. Future research can also investigate differences between adults and children, which lead to the occurrence of SSI among children. Another opportunity for future research is to investigate which risk factors are predictive for the occurrence of SSI over different periods. Doing this will enable healthcare workers to identify which risk factors explain the occurrence of SSI soon after surgery, towards the end of the 30 days and even later for implantation surgeries. These insights can help set guidelines to determine the vigilance necessary to mitigate the risk of SSI on a patient level.

## Conclusion

This study shows that data-driven cut-offs can be used to identify risk factors which would not have been identified by only using standard medical cut-offs. Preoperative temperature and antibiotic use were identified as risk factors for digestive, orthopaedic, thoracic system surgeries, while the duration of surgery and age were identified as risk factors for orthopaedic and thoracic system surgeries, respectively. In contrast with literature, this study found that an SSI is more likely to occur in children (age < 18) than in adults after thoracic system surgeries. Statistical modelling has been important to quantify important risk factors and indicate their significance. Clinical studies using retrospective data are important to carry out, despite limitations in the data sets. To this end, future studies should use both standard medical cut-offs and data-driven cut-offs to investigate risk factors.

## Supporting information

**S1 Table. Risk factors identified from multivariate analysis during literature search.** (DOCX)

**S1 Formulae. The multivariate logistic regression equations based on the data-driven cut-offs.** (DOCX)

## Acknowledgments

We would like to thank C. P. (Conrad) van der Hoeven, A.G.D (Arnim) Mulder and M. (Marius) Vogel for their help and constant willingness to help with questions regarding the data used for this study. Also, thank you to R. H. (Roel) Streefkerk for the work done to organise and combine the data as well as producing the SSI outcome variable used in this study.

## Author Contributions

**Conceptualization:** J. M. van Niekerk, M. C. Vos, A. Stein, L. M. A. Braakman-Jansen, A. F. Voor in 't holt, J. E. W. C. van Gemert-Pijnen.

**Data curation:** J. M. van Niekerk, M. C. Vos.

**Formal analysis:** J. M. van Niekerk, A. F. Voor in 't holt.

**Investigation:** J. M. van Niekerk, M. C. Vos, A. F. Voor in 't holt.

**Methodology:** J. M. van Niekerk, M. C. Vos, A. Stein, A. F. Voor in 't holt.

**Supervision:** A. Stein, L. M. A. Braakman-Jansen, A. F. Voor in 't holt, J. E. W. C. van Gemert-Pijnen.

**Validation:** J. M. van Niekerk.

**Writing – original draft:** J. M. van Niekerk, M. C. Vos, A. Stein, L. M. A. Braakman-Jansen, A. F. Voor in 't holt.

**Writing – review & editing:** J. M. van Niekerk, M. C. Vos, A. Stein, L. M. A. Braakman-Jansen, A. F. Voor in 't holt, J. E. W. C. van Gemert-Pijnen.

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
