## [Decision Letter · Decision Letter 0]

16 Jun 2020

PONE-D-20-14495

Risk factors for surgical site infections using a data-driven approach

PLOS ONE

Dear Dr. van Niekerk,

Thank you for submitting your manuscript to PLOS ONE. After careful consideration, we feel that it has merit but does not fully meet PLOS ONE’s publication criteria as it currently stands. Therefore, we invite you to submit a revised version of the manuscript that addresses the points raised during the review process.

We look forward to receiving your revised manuscript.

Kind regards,

Francesco Di Gennaro

Academic Editor

PLOS ONE

Journal Requirements:

2. We note that you have provided information about your ethics approval and data anonymization in the Ethics Statement. We ask that you additionally provide this information in your Methods section.

Reviewers' comments:

Reviewer's Responses to Questions

**Comments to the Author**

1. Is the manuscript technically sound, and do the data support the conclusions?

Reviewer #1: Yes

Reviewer #2: Yes

2. Has the statistical analysis been performed appropriately and rigorously? 

Reviewer #1: Yes

Reviewer #2: Yes

3. Have the authors made all data underlying the findings in their manuscript fully available?

Reviewer #1: Yes

Reviewer #2: Yes

4. Is the manuscript presented in an intelligible fashion and written in standard English?

Reviewer #1: Yes

Reviewer #2: Yes

5. Review Comments to the Author

Reviewer #1: In this work, Authors conducted a robust data-driven approach to identify risk factors for surgical site infections after digestive system, orthopedic or thoracic surgery procedures. Overall, the work is well written and clear. The statistical analysis is appropriate and well performed.

The major limitation of this work is the lack of certain information that could have influenced the final result of multivariable analysis.

However, this work is worth for publication after few minor revisions

Introduction

1) Lines 48-49: Many research have been conducted until now in order to identify risk factors for SSI. However, one important point often overlooked is the importance of clinical setting. Indeed, risk factors for SSI could importantly differ between low income countries and high income countries.

A recent interesting work addressed this topic (Di Gennaro F, et al. Maternal caesarean section infection (MACSI) in Sierra Leone: a case-control study. Epidemiol Infect . 2020 Feb 27;148:e40. doi: 10.1017/S0950268820000370.) and could be cited, along with a brief paragraph discussing this point.

Discussion

1) The antibiotic use as a risk factor for SSI is interesting but controversial. In fact, although many guidelines suggest to limit the use of the antibiotic prophylaxis at the day of the surgery, often antibiotics are prescribed several days before and after the surgical day.

In my opinion Authors should give a possible explanation for this very important result possibly based on their data.

2) Did any data regarding the reason for surgical procedures were available? It is widely accepted that the cause of surgical procedure (infection, cancer, prosthetic implantation, etc) and the general clinical condition of patient (immunocompromised, severely undernourished, etc) are significant predictors of SSI. If these information are not available, this point should be included among limitations.

Reviewer #2: In this manuscript Johan Magnus van Niekerk and co-authors tried to identify major risk factors for surgical site infections from digestive, thoracic and orthopaedic system surgeries. To reach this aim they previously identified some known risk factors in literature and then they compared them with data extrapolated from a database of a large tertiary care hospital in the Netherlands. This study could give an important contribution on prevention of surgical site infection by identifying modifiable risk factors and subsequently preventative actions. The title and abstract are appropriate for the content of the text. There are my comments sorted by section.

Methods:

• line 93: I think that could be useful to clarify how the nosocomial infection index was obtained, in particular which parameters where included to determine it.

• Line 192-205: I think that this paragraph should be removed. It appears too technical and probably not necessary for the purpose of the article.

Discussion:

• Lines 240-241: when you talk about antibiotic usage before surgery do you refer to their use in a broad sense or there are particular class of antibiotics related to a major risk? It could be interesting to examine which classes are related to a major risk for SSI.

• Lines 256-257: there is a repetition, you just defined the general risk factors in lines 225-228

6. PLOS authors have the option to publish the peer review history of their article (what does this mean?). If published, this will include your full peer review and any attached files.

Reviewer #1: No

Reviewer #2: No

---

## [Author Response · Author response to Decision Letter 0]

21 Sep 2020

Response to reviewers – PONE-D-20-14495

Risk factors for surgical site infections using a data-driven approach

Johan Magnus van Niekerk, Margreet C. Vos, Alfred Stein, Annemarie Braakman-Jansen, Anne F. Voor in ’t holt, Lisette van Gemert-Pijnen

We are grateful for your valuable comments which helped us to improve our manuscript. We sincerely appreciate the time you spent reviewing this manuscript. We revised our manuscript in accordance with your comments. Please find our responses to the comments below.

Reviewer #1: 

Introduction

1. Lines 48-49: Many research have been conducted until now in order to identify risk factors for SSI. However, one important point often overlooked is the importance of clinical setting. Indeed, risk factors for SSI could importantly differ between low income countries and high income countries.

A recent interesting work addressed this topic (Di Gennaro F, et al. Maternal caesarean section infection (MACSI) in Sierra Leone: a case-control study. Epidemiol Infect . 2020 Feb 27;148:e40. doi: 10.1017/S0950268820000370.) and could be cited, along with a brief paragraph discussing this point.

We welcome this suggestion and consider the introduction as more rounded because of this addition. The following text was added to the introduction, and the suggested research article cited: 

L51 – L53: “Risk factors in low-income countries also include unemployment and level of education due to the disparity in socioeconomic status [20].”

Discussion

1. The antibiotic use as a risk factor for SSI is interesting but controversial. In fact, although many guidelines suggest to limit the use of the antibiotic prophylaxis at the day of the surgery, often antibiotics are prescribed several days before and after the surgical day.

In my opinion Authors should give a possible explanation for this very important result possibly based on their data.

Although the data regarding the timing and administration of surgical antibiotic prophylaxis were not available for this study, the reviewer’s comment led us to further investigate the relationship between the occurrence of SSI and the time between J01 antibiotics prescription time and surgery. The occurrence of SSI seems to vary for different times between antibiotics prescription and surgery (Fig 1), but hypotheses regarding this relationship should be evaluated using case-control studies with specific data regarding the reason for the prescription, the type of antimicrobials and the timing of the administration thereof.

Fig 1. The occurrence of SSI for the time between the J01 antibiotics prescription start time and the start time of the surgery by decile.

m, minutes; h, hours; d, days; #SSI, number of SSI occurrences; %SSI, percentage of SSI occurrences.

1The vertical axis starts at 75% to increase visibility.

2The horizontal axis stipulates the endpoints of the respective deciles of the distribution of the time between prescription and surgery.

We added the following text to the discussion section to further share our insight with the readers:

L280 – L283: “Systemic antibiotics are typically prescribed to stabilise patients before undergoing surgery. A possible explanation for the increased occurrence of SSI associated with antimicrobials prescribed before surgery could be that these patients were not completely stabilised before surgery which increased their risk of SSI.”

2. Did any data regarding the reason for surgical procedures were available? It is widely accepted that the cause of surgical procedure (infection, cancer, prosthetic implantation, etc) and the general clinical condition of patient (immunocompromised, severely undernourished, etc) are significant predictors of SSI. If these information are not available, this point should be included among limitations.

We were hoping to include this information in the study but soon realised that these data were only available in free text field information, and there was no way to extract these data efficiently.

We added the following sentence to the Limitations section: 

L288 – L290: “Surgeries were aggregated into three broad groups of surgical procedures which serve as a proxy for the reason for surgery but leads to the loss of information regarding the exact reasons for the surgery.”

L294-295: “A comprehensive list of variables related to the nutritional and immunological alterations of the patients was not included in the analyses as they were not available from the data.”

Reviewer #2: 

Methods

1. line 93: I think that could be useful to clarify how the nosocomial infection index was obtained, in particular which parameters where included to determine it.

Thank you for bringing this to our attention. The references to the two articles which describe the nosocomial infection index, used as outcome variables in our study, are now provided immediately after the algorithm is mentioned for the first time. We also now refer to Streefkerk et al. in the text to make this referral clearer.

2. Line 192-205: I think that this paragraph should be removed. It appears too technical and probably not necessary for the purpose of the article.

This paragraph referred to was added as supplementary material (S2 Formulae) instead.

Discussion

1. Lines 240-241: when you talk about antibiotic usage before surgery do you refer to their use in a broad sense or there are particular class of antibiotics related to a major risk? It could be interesting to examine which classes are related to a major risk for SSI.

Antibiotics in J01 Anatomical Therapeutic Chemical were used this study. This definition is specified in the variable definition table, but the legend was not complete. We agree with the reviewer that delving deeper into the subclasses of antibiotics is of great importance, although not in the scope of this study. 

The following changes were made to the manuscript:

L101-103: The following text was added to the methods section where the data are described: “Data regarding the prescription of antimicrobials, in the J01 class of the Anatomical Therapeutic Chemical (ATC) classification system [28], were also included.”

L161-L162: The following text was to the legend of the variable definition table: “ATC, Anatomical Therapeutic Chemical; WHO, World Health Organization.”

2. Lines 256-257: there is a repetition, you just defined the general risk factors in lines 225-228

Thank you, we have removed the repetition.

---

## [Decision Letter · Decision Letter 1]

7 Oct 2020

Risk factors for surgical site infections using a data-driven approach

PONE-D-20-14495R1

Dear Dr. Johan,

We’re pleased to inform you that your manuscript has been judged scientifically suitable for publication and will be formally accepted for publication once it meets all outstanding technical requirements.

Kind regards,

Francesco Di Gennaro

Academic Editor

PLOS ONE

Additional Editor Comments (optional):

Dear Authors congratulations!

Reviewers' comments:

Reviewer's Responses to Questions

**Comments to the Author**

1. If the authors have adequately addressed your comments raised in a previous round of review and you feel that this manuscript is now acceptable for publication, you may indicate that here to bypass the “Comments to the Author” section, enter your conflict of interest statement in the “Confidential to Editor” section, and submit your "Accept" recommendation.

Reviewer #1: All comments have been addressed

Reviewer #2: All comments have been addressed

2. Is the manuscript technically sound, and do the data support the conclusions?

Reviewer #1: Yes

Reviewer #2: Yes

3. Has the statistical analysis been performed appropriately and rigorously? 

Reviewer #1: Yes

Reviewer #2: Yes

4. Have the authors made all data underlying the findings in their manuscript fully available?

Reviewer #1: Yes

Reviewer #2: Yes

5. Is the manuscript presented in an intelligible fashion and written in standard English?

Reviewer #1: Yes

Reviewer #2: Yes

6. Review Comments to the Author

Reviewer #1: (No Response)

Reviewer #2: (No Response)

7. PLOS authors have the option to publish the peer review history of their article (what does this mean?). If published, this will include your full peer review and any attached files.

Reviewer #1: No

Reviewer #2: **Yes: **Mariani Michele Fabiano

---

## [Editor Report · Acceptance letter]

9 Oct 2020

PONE-D-20-14495R1 

Risk factors for surgical site infections using a data-driven approach 

Dear Dr. van Niekerk:

I'm pleased to inform you that your manuscript has been deemed suitable for publication in PLOS ONE. Congratulations! Your manuscript is now with our production department. 

Kind regards, 

on behalf of

Dr. Francesco Di Gennaro 

Academic Editor

PLOS ONE